# Characterization of anti-canine CD20 antibody 4E1-7-B_f and comparison with commercially available anti-human CD20 antibodies

Takuya Mizuno[1,2,3]*, Yukinari Kato[4], Toshihiro Tsukui[5], Masaya Igase[1,2,3]

1 Laboratory of Molecular Diagnostics and Therapeutics, Joint Graduate School of Veterinary Medicine, Yamaguchi University, Yamaguchi, Japan, 2 Division of Translational Research for One Medicine, Research Institute for Cell Design Medical Science, Yamaguchi University, Yamaguchi, Japan, 3 Japan Small Animal Cancer Center, Affiliated with the Japan Small Animal Medical Center Foundation, Tokorozawa, Saitama, Japan, 4 Department of Antibody Drug Development, Tohoku University Graduate School of Medicine, Sendai, Miyagi, Japan, 5 Nippon Zenyaku Kogyo Co., Ltd., Koriyama, Fukushima, Japan

* mizutaku@yamaguchi-u.ac.jp

## Abstract

This study characterizes the previously reported anti-canine CD20 antibody 4E1-7-B_f and compares this with commercially available anti-human CD20 antibodies, rituximab and an obinutuzumab biosimilar. While the obinutuzumab biosimilar exhibited binding to canine CD20 in a CD20-transduced cell line, canine B-cell lymphoma cell line (CLBL-1/luc), and canine CD21 + B cells from healthy dogs, functional assays revealed the superiority of 4E1-7-B_f in antibody-dependent cellular cytotoxicity and complement-dependent cytotoxicity activities over those of the obinutuzumab biosimilar. Epitope analysis suggested an extracellular region on canine CD20 targeted by 4E1-7-B_f. Furthermore, the lipid raft localization of CD20 in CLBL-1/luc cells by treatment with 4E1-7-B_f classified this antibody as a type II anti-CD20 antibody which works with strong ADCC activity, similar to the obinutuzumab biosimilar, unlike rituximab, a type I anti-CD20 antibody, whose main action is CDC activity. These findings underscore the potential clinical utility of 4E1-7-B_f, emphasizing the specificity, potency, and therapeutic promise in canine lymphoma treatment.

## Introduction

The anti-human CD20 antibody rituximab is one of the most popular antibody drugs in human medicine and has been used for treating B-cell type lymphoma and autoimmune diseases for more than two decades. The clinical efficacy of rituximab is obtained by killing CD20-positive B cells, and many types of anti-CD20 antibody drugs, such as ofatumumab [1], obinutuzumab [2], [90]Y ibritumomab tiuxetan [3], and [131]I tositumomab [4], with various additional functional features, have been developed to achieve more potent clinical effects. Anti-CD20 antibodies are cytotoxic antibody

**Data availability statement:** All relevant data are within the paper and its Supporting Information files.

**Funding:** This study was financially supported by Nippon Zenyaku Kogyo Co., Ltd. in the form of a collaborative research fund received by TM. This study was also financially supported by the Japan Agency for Medical Research and Development (AMED) in the form of a grant (JP22ama121008) received by YK. The funders had no role in study design, data collection and analysis, decision to publish, or preparation of the manuscript.

**Competing interests:** The authors have read the journal's policy and have the following competing interests: TT is an employee of Nippon Zenyaku Kogyo Co., Ltd. There are no patents, products in development or marketed products associated with this research to declare. This does not alter our adherence to PLOS ONE policies on sharing data and materials.

drugs exerting antitumor effects through antibody-dependent cell cytotoxicity (ADCC), complement-dependent cytotoxicity (CDC), and direct cytotoxic activity, but these functions vary widely among the therapeutic agents directed towards this target. Anti-CD20 antibodies such as rituximab and ofatumumab were classified as type I antibodies [5], induce localization of CD20 to the lipid raft, and have high CDC-induction ability because of biomolecular binding to CD20, whereas antibodies such as obinutuzumab, were classified as type II antibodies, do not induce localization of CD20 to the lipid raft and have weak CDC activity, although direct cell death and ADCC and ADCP activities are also strongly observed [6]. In addition, obinutuzumab has enhanced ADCC activity through modification of the sugar chain in the Fc region of the antibody [7].

Canine B-cell lymphoma is one of the most common tumors occurred in dogs and is primarily treated with multiagent chemotherapies, although their efficacies are limited [8]. Despite initial response rates of approximately 80–90% with CHOP-based chemotherapy protocols, most dogs experience relapse within 6–12 months, and the long-term remission rate remains low, with median survival times of only 12–14 months. Additionally, approximately 20–40% of dogs do not respond to rescue chemotherapy following relapse, highlighting the urgent need for novel therapeutic strategies. As with human B-cell lymphoma, treatment with the anti-CD20 antibody drug has long shown promise. However, since a previous report showed that rituximab did not react with canine CD20 [9], several monoclonal antibodies against canine CD20 have been developed [10–14], although no anti-canine CD20 antibody drug is commercially available. Furthermore, the mode or mechanism of action of monoclonal antibodies that have been reported has not yet been analyzed in detail.

We previously established an anti-dog CD20 monoclonal antibody, 4E1-7, and generated a chimeric antibody, 4E1-7-B, based on this molecule [11]. We also developed a fucose-deficient variation of this chimeric antibody, 4E1-7-B_f antibody, which showed significant cytotoxicity activity *in vitro* as compared with original 4E1-7 or 4E1-7-B antibodies. We further showed that this antibody efficiently depleted B cells in the peripheral blood in healthy beagle dogs. We are currently conducting clinical trials using 4E1-7-B_f on dogs with B-cell lymphoma. However, a more detailed analysis of the 4E1-7-B_f antibody is required to understand the function of this antibody.

In this study, we investigated whether 4E1-7-B_f antibodies can cross-react with CD20 in other species such as humans, cats, and cattle. We also examined the CD20 epitope of the 4E1-7-B_f antibody based on those of rituximab and obinutuzumab, which are representative of type I and type II antibodies, respectively, and examined whether an obinutuzumab biosimilar can recognize CD20 in dogs. Finally, we determined whether the 4E1-7-B_f antibody was a type I or II group antibody based on the CD20 localization into lipid raft.

## Materials and methods

### Cells

The rat kidney cell line, NRK, and the retroviral packaging cell line, PLAT-gp, were cultured in D10 complete medium (Dulbecco's modified Eagle medium supplemented

with 10% fetal bovine serum [FBS], 100 U/mL penicillin, 100 µg/mL streptomycin, and 55 µM 2-mercaptoethanol). Human B-cell line, Ramos, canine B-cell lymphoma cell line, CLBL-1/luc, and A20/luc/cCD20, target cell line for ADCC [15] were cultured in R10 complete medium (RPMI1640 supplemented with same additives in D10 complete medium). PLAT-gp and CLBL-1 cell lines were kindly provided by Dr. Toshio Kitamura and Dr. Barbara Rütgen, respectively. As effector cells for ADCC assay, NK-92/cCD16γ cells were used in R10 complete medium containing 20% FBS plus 1000 IU/mL human IL-2 (Proleukin®; Chiron Therapeutics, Emeryville, CA, USA). Peripheral blood mononuclear cells (PBMCs) were isolated from three healthy beagles, which were kept for blood donors in Yamaguchi University Animal Medical Center as previously described [11].

## Anti-CD20 antibodies

Defucosylated anti-canine CD20 chimeric antibody (4E1-7-B_f) was reported in our previous study [11]. The anti-human CD20 antibody, rituximab (Chugai Pharmaceutical Co., Ltd., Tokyo, Japan) and anti-hCD20-Ga-hIgG1fut (InvivoGen, San Diego, CA), a biosimilar obinutuzumab, were used as a type I and II antibodies, respectively.

## Retroviral expression plasmids

Bovine CD20 and feline CD20 were amplified via PCR with primers YTM2088 (5′- AC<u>GGATCC</u>ATGACGACACCCAG-GAATTCAATG −3′; underlined: BamHI) and YTM2089 (5′- <u>GTCGATGTCATGATCTTTATAATC</u> AGGGACACTGTCGTTC 3′; DYKDHDID tag underlined), and primers, YTM2090 (5′-AC<u>GGATCC</u> ATGACAACACCCAGAAATTCAATG-3′; underlined: BamHI) and YTM2091 (5′- <u>GTCGATGTCATGATCTTTATAATC</u> AGGAATGCTATCGTTTTCTATCAG-3′; DYKDHDID tag underlined), using bovine PBMCs and feline thymus cDNAs as a template, respectively. Both products were amplified again with primers YTM2088/YTM838 and YTM2090/YTM838 to tag the expression plasmids of bovine and feline CD20 with FLAG, respectively. Both products were cut with BamHI and ligated at BamHI and SnaBI sites of pMXs-IP, producing pMx-IP-bCD20-FL#3 and pMx-IP-fCD20-FL#2, respectively. Nucleotide sequences of both plasmids were confirmed via Sanger sequencing.

## Stable cell lines

Bovine CD20 and feline CD20 overexpressing cell lines, NRK/bCD20-FL and NRK/fCD20-FL were prepared as for NRK/cCD20-FL, which was previously established [11]. Briefly, pMx-IP-bCD20-FL#3 and pMx-IP-fCD20-FL#2 were transfected with pCVSVG into the PLAT-gp cell line, and the retroviral supernatants were used to infect the NRK cell line. The resultant cell lines were NRK/bCD20-FL and NRK/fCD20-FL.

## Flow cytometry

A total of $2 \times 10^5$ cells were washed with FACS buffer (PBS with 2% FBS and 0.1% NaN3) and incubated with primary antibodies; anti-canine CD20 antibody, 4E1-7-B_f, anti-human CD20 antibodies, rituximab, and anti-hCD20-Ga-hIgG1fut (obinuzutumab biosimilar), and PE-conjugated anti-canine CD21 antibody (RRID: AB_323238). ChromPure dog IgG, whole molecule (Jackson ImmunoResearch Laboratories Inc., West Grove, PA; RRID: AB_2336981) and human IgG$_1$ kappa isotype control antibody (GeneTex, Inc., Irvine, CA; RRID: AB_10634045) are used as an isotype control. After incubation at 4°C for 30 min, cells were washed, followed by staining with secondary antibodies: Alexa647-conjugated anti-dog IgG (Jackson ImmunoResearch Laboratories Inc.; RRID: AB_2339372), Dylight649-conjugated anti-rat IgG (BioLegend Japan, Tokyo, Japan; RRID: AB_2562967), or Alexa Fluor 647-conjugated anti-human IgG (BioLegend Japan; RRID: AB_2728443). After staining with propidium iodide, propidium iodide negative cells were gated in, and then stained cells were analyzed using CytoFLEX (Beckman Coulter Inc., Tokyo, Japan), followed by analysis with FlowJo software (TreeStar, Woodbum, OR). The gating strategy used for staining PBMCs is shown in S1 Fig in S1 File. $K_D$ value as a binding affinity was determined by flow cytometry using serially diluted antibody, as described in our previous study [11].

## ADCC and CDC assay

The ADCC assay was performed using A20/luc/cCD20 and NK-92/cCD16γ cell lines as target and effector cells, according to our previously established ADCC system [15]. The CDC assay was performed using CLBL-1/luc cells as the target cell line as previously described [11].

**Lipid raft translocation analysis.** Lipid raft translocation analysis of CD20 after binding of antibody was performed as previously described [16]. Briefly, $1 \times 10^6$ of Ramos and CLBL-1/luc cells were treated with the isotype control, 4E1-7-B_f, rituximab, or obinutuzumab similar for 5 min. Stimulated cells were collected, washed with phosphate-buffered saline, and lysed in 50 µL of lysis buffer (1% Triton X-100, 50 mM Tris-HCl, 150 mM NaCl, pH7.5, 1 mM EDTA, 1 mM $Na_3VO_4$, and Protease Inhibitor Cocktail [Nakarai Tesque, Inc., Kyoto, Japan]) at 4°C for 30 min. After centrifugation, supernatants and cell pellets were collected as soluble and insoluble fractions, respectively, for western blotting analysis.

## Western blotting

For detection of overexpressed proteins, cells were lysed in 1% NP40 lysis buffer, followed by centrifugation. Supernatants were collected and used as cell lysates. Cell lysates, soluble fractions, or insoluble fractions were used for SDS-PAGE, followed by western blotting as described previously [11]. Primary antibodies were as follows: monoclonal anti-Flag M2 antibody (Sigma-Aldrich Crop., St. Louis MO; RRID: AB_26044), anti-β-actin antibody, mouse monoclonal (clone AC-15, Sigma-Aldrich Crop.; RRID: AB_476692), anti-CD71 monoclonal antibody (clone H68.4, Santa Cruz Biotechnology, Dallas, TX; RRID: AB_1120670), anti-CD20 polyclonal antibody (Thermo Fisher Scientific, Fremont, CA, USA; RRID: AB_10980806), and anti-Lyn polyclonal antibody (clone 44, Santa Cruz Biotechnology; RRID: AB_2281450). The horseradish peroxidase (HRP) –conjugated goat anti-mouse IgG secondary antibody (BioLegend Japan; RRID: AB_15009) and HRP–conjugated goat anti-rabbit IgG antibody (Jackson ImmunoResearch Laboratories Inc.; RRID: AB_2307391) were used as secondary antibodies. The membrane was visualized via immersion in Western Lightning chemiluminescence reagent (Perkin Elmer, Foster City, CA, USA). Immunoreactive bands were visualized using the Amersham Image Quant 800 (Cytiva, Tokyo, Japan).

## Statistical analysis

Tukey Kramer test was conducted to perform multiple comparison. P-values less than 0.05 was considered to be statistical significance. All statistical tests were performed using JMP Pro software version 16 (SAS Institute Japan).

## Results

### Crossreactivity of anti-CD20 antibodies

We first used flow cytometry to examine the cross-reactivity of our previously established defucoyslated rat-canine chimeric anti-canine CD20 antibody, 4E1-7-B_f, to CD20 molecules of other species (Fig 1A). The 4E1-7-B_f antibody bound to canine CD20 exogenously overexpressed in NRK cells and in CLBL-1/luc cells, a canine B-cell lymphoma cell line endogenously overexpressing CD20. However, neither bovine or feline CD20 were detected by 4E1-7-B_f in overexpressed NRK cells, and human CD20 was also not detected by 4E1-7-B_f in Ramos cells endogenously expressing human CD20. This result indicates that 4E1-7-B_f is specific to canine CD20.

Conversely, anti-human CD20 antibodies, rituximab and obinutuzumab biosimilar bound to canine CD20 overexpressed in NRK cells, although the binding intensity was lower than that of 4E1-7-B_f ($K_D$ value, $17.42 \pm 1.76 \times 10^{-9}$ M vs $4.2 \pm 2.6 \times 10^{-9}$ M). Endogenously expressed canine CD20 in CLBL-1/luc was not detected by rituximab but was weakly detected by the obinutuzumab biosimilar. This result indicated that rituximab and the obinutuzumab biosimilar bound to canine CD20, but not as strongly as 4E1-7-B_f. Both 4E1-7-B_f and the obinutuzumab biosimilar dose-dependently bound to canine CD20 (Fig 1B), although the binding of 4E1-7-B_f was much stronger than that of the obinutuzumab biosimilar.

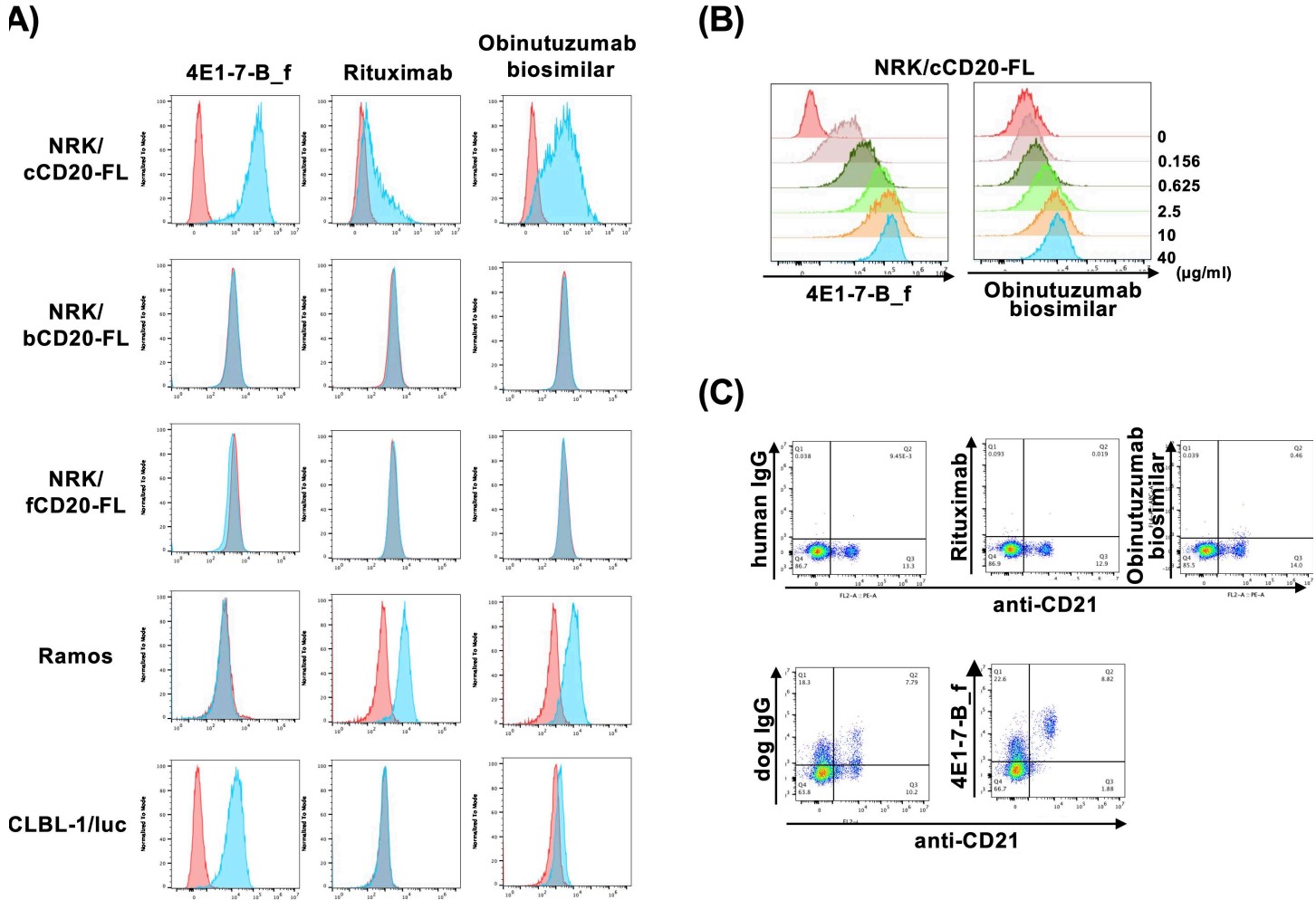

**Fig 1. Reactivity of anti-human CD20 antibodies to canine CD20.** (a) NRK cell lines overexpressing canine CD20 (NRK/cCD20-FL), bovine CD20 (NRK/bCD20-FL), feline CD20 (NRK/fCD20-FL), human B-cell line, Ramos, and canine B-cell line (CLBL-1/luc) were stained with isotype control (histogram in red), 4E1-7-B_f, rituximab, and obinutuzumab biosimilar (histogram in blue), followed by appropriate secondary antibodies. Stained cells were analyzed via flow cytometry. (b) Obinuzutumab biosimilar bound to canine CD20 overexpressed in NRK cells. NRK/cCD20-FL was stained by serially diluted amounts of 4E1-7-B_f or obinutuzumab biosimilar; 0 µg/mL indicates isotype control. (c) Rituximab and obinutuzumab biosimilar did not bind to canine CD21 + B cells. Canine PBMCs were isolated from three dogs and stained with anti-canine CD21 antibody and rituximab, obinutuzumab biosimilar, or 4E1-7-B_f. Human IgG and dog IgG were used as isotype controls. Gating strategy is shown in S1 Fig in S1 File.

We also tested the binding of these antibodies to CD20 endogenously expressed in canine CD21-positive B-cell fraction in PBMCs from healthy beagles. 4E1-7-B_f bound to CD21 + B cells, but rituximab did not, and the obinutuzumab biosimilar only bound weakly (Fig 1C).

## Epitope analysis of 4E1-7-B_f

The amino acid sequences of CD20 of species are compared in Fig 2A while a structural schematic of canine and human CD20s and epitopes of rituximab, obinutuzumab, and ofatumumab is shown in Fig 2b [17]. We applied this information and our flow cytometry analysis (Fig 1) to determine the epitope of 4E1-7-B_f using canine CD20 mutants. Since 4E1-7-B_f did not bound to human CD20 (Fig 1A), several amino acids in canine CD20 were substituted to corresponding amino

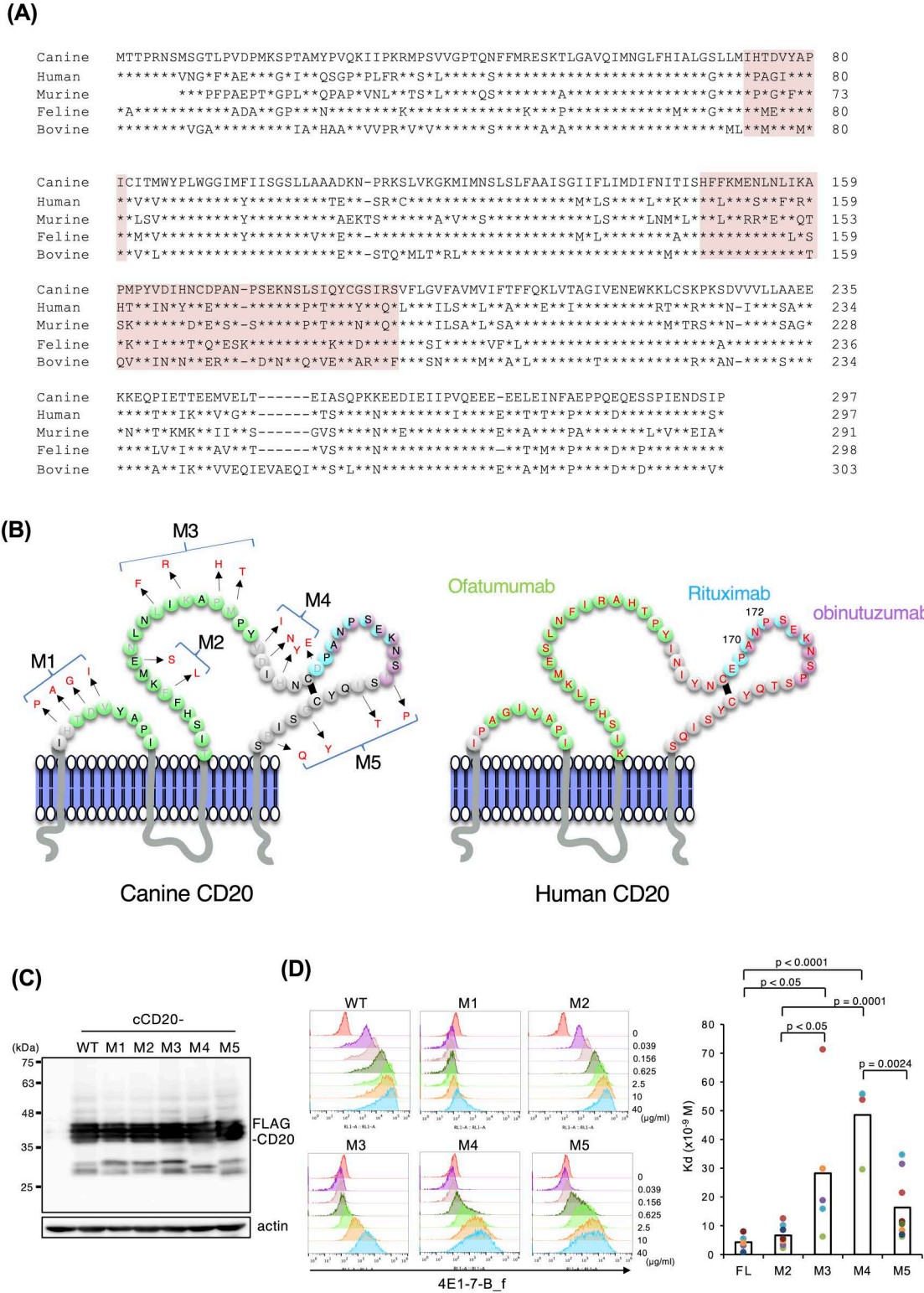

**(A)**

```
Canine    MTTPRNSMSGTLPVDPMKSPTAMYPVQKIIPKRMPSVVGPTQNFFMRESKTLGAVQIMNGLFHIALGSLLMIHTDVYAP    80
Human     *******VNG*F*AE***G*I**QSGP*PLFR**S*L*****S****************************G***PAGI***    80
Murine       ***PFPAEPT*GPL**QPAP*VNL**TS*L***QS******A***********G***P*G*F**                73
Feline    *A**********ADA*GP***N********K*********K***P************M***G****ME****                80
Bovine    ********VGA*********IA*HAA**VVPR*V*V******S*****A*A****************ML**M***M*            80

Canine    ICITMWYPLWGGIMFIISGSLLAAADKN-PRKSLVKGKMIMNSLSLFAAISGIIFLIMDIFNITISHFFKMENLNLIKA    159
Human     **V*V*********Y*********TE**-SR*C*****************M*LS****L**K****L**S**F*R*            159
Murine    **LSV*********Y********AEKTS******A*V**S***********LS***LNM*L**L**RR*E**QT            153
Feline    **M*V********Y*******V**E**-****************M*L********A***********L*S                159
Bovine    **V*L****************E**-STQ*MLT*RL***************M***********T                        159

Canine    PMPYVDIHNCDPAN-PSEKNSLSIQYCGSIRSVFLGVFAVMVIFTFFQKLVTAGIVENEWKKLCSKPKSDVVVLLAAEE    235
Human     HT**IN*Y*E***-*****P*T***Y**Q*L***ILS**L**A***E*I********RT**R***N-I***SA**            234
Murine    SK******D*E*S*-S*****P*T***N**Q*****ILSA*L*SA*************M*TRS**N-****SAG*            228
Feline    *K**I***T*Q*ESK********K**D********SI*****VF*L****************A*********            236
Bovine    QV**IN*N**ER**-*D*N**Q*VE**AR**F***SN****M**A*L*******T************R**AN-****S***      234

Canine    KKEQPIETTEEMVELT------EIASQPKKEEDIEIIPVQEEE-EELEINFAEPPQEQESSPIENDSIP                297
Human     ****T**IK**V*G**------*TS****N********I****E**T*T**P****D**********S*                297
Murine    *N**T*KMK**II**S------GVS****N**E*********E**A****PA*******L*V**EIA*                  291
Feline    ****LV*I***AV**T------*VS****N***********-**T*M**P****D**P**********                  298
Bovine    ****A**IK**VVEQIEVAEQI**S*L**N***********E**A*M**P****D**D*******V*                  303
```

**(B)**

**(C)**

**(D)**

**Fig 2. Epitope analysis of 4E1-7-B_f antibody.** (a) Comparison of amino acid sequences of canine, human, murine, feline, and bovine CD20. * indicates the identical amino acid. Shaded boxes in pink indicate the extracellular domains of CD20 of each species. (b) Comparison of schematic structures of canine and human CD20 molecules. Green, light blue, and purple amino acid regions indicate the epitopes of ofatumumab, rituximab, and

obinutuzumab, respectively. Amino acids in red in dog CD20 represent amino acids substituted into human CD20, and M1 to M5 indicate that only that portion of canine CD20 was replaced by human CD20 amino acids. (c) Expression of mutant forms of cCD20 (M1 to M5) in NRK cell lines as assessed via western blotting analysis. WT, cCD20-M1, M2, M3, M4, and M5 were stably expressed in NRK cells. Whole cell lysates were extracted from trans-duced cells and analyzed via western blotting with an anti-FLAG antibody; actin was used as a loading control. For transparency, the full-length images of the cropped western blots are shown in S2 Fig in S1 File. Representative results from two independent experiments are shown. Replicates of the full dataset are provided in S3 Fig in S1 File. (d) Expression of mutant forms of cCD20 (M1-M5) in NRKs via flow cytometry and comparison of the dissoci-ation constant (Kd) values of an antibody across five different cell types (FL, M2, M3, M4, M5). Cells were stained with 4E1-7-B_f antibody at concen-trations prepared by a 4-fold serial dilution, starting from 40 μg/ml down to 0.04 μg/ml, followed by a secondary antibody. Stained cells were analyzed by flow cytometry. One representative histogram results of several independent experiments is shown on the left side. Individual colored dots in right graph represent independent experimental measurements of Kd value, and bars indicate the mean Kd values for each cell type. Statistical analysis was performed using the Tukey Kramer multiple comparison analysis, with significant differences indicated by p-values. Significant differences were observed among multiple groups, as shown in the figure.

acids in human CD20 (from M1 to M5) as shown in Fig 2B. cCD20-M1, M2, M3, M4, and M5 contain H74P, T75A, D76G and V77I substitutions, F149L and E152S substitutions, L156F, K158R, P160H, and M161T substitutions, V164I, D165N, H167Y, and D170E substitutions, L180P, I182T, G186Y, and R189Q substitutions, respectively. Each mutant was stably transduced into NRK cells and was expressed equally in each transduced cell line as shown via western blotting (Fig 2C). Flow cytometric analysis of NRK cells transduced with these mutants (M1 to M5) by 4E1-7-B_f was performed (Fig 2D). Binding of 4E1-7-B_f was completely abrogated in NRK/cCD20-M1 cells but was unchanged in NRK/cCD20-M2 cells ($K_D$ value, $6.6 \pm 3.8 \times 10^{-9}$ M), as compared with NRK/cCD20/WT cells ($K_D$ $4.2 \pm 2.6 \times 10^{-9}$ M). Unexpectedly, NRK/cCD20-M3 ($K_D$ value, $28.3 \pm 9.7 \times 10^{-9}$ M), and NRK/cCD20-M4 ($K_D$ value, $48.6 \pm 15.2 \times 10^{-9}$ M) showed a significantly less binding of 4E1-7-B_f than NRK/cCD20/WT cells ($p < 0.05$), and NRK/cCD20-M5 cells ($K_D$ value, $16.3 \pm 12.1 \times 10^{-9}$ M) showed slightly higher affinity than NRK/cCD20-M4 cells.

### Comparison of ADCC and CDC activity of 4E1-7-B_f and obinutuzumab biosimilar

Next, we compared the ADCC and CDC effects of the obinutuzumab biosimilar with 4E1-7-B_f. We previously showed that 4E1-7-B_f induced profound dose-dependent ADCC effects on A20/cCD20/luc cells, as well as CDC effects on CLBL-1/luc cells at 10 μg/mL [11], whereas the obinutuzumab biosimilar produced ADCC effects that were approximately 100-fold less than those of 4E1-7-B_f and had no CDC activity (Fig 3). These results were consistent with the flow cyto-metric analysis.

### Lipid raft translocation of CD20 in the cells treated with 4E1-7-B_f and obinutuzumab biosimilar

Finally, to clarify the type of 4E1-7-B_f antibody, raft localization of CD20 molecules after binding of antibody was exam-ined. Rituximab and the obinutuzumab biosimilar were used as controls of type I and II antibodies, respectively. As shown in Fig 4, treatment of Ramos cells with rituximab moved CD20 to lipid raft fraction but this did not occur with treatment with the obinutuzumab biosimilar. Treatment of CLBL-1/luc cells with 4E1-7-B_f showed no movement of CD20 in the lipid raft fraction. This result showed that 4E1-7-B_f exhibited characteristics of a type II antibody.

### Discussion

It is crucial to define the mode of action of different cytotoxic antibodies to design their further development. We previously reported 4E1-7-B_f as an antibody drug with strong cytotoxic activity *in vitro* [11], although the epitope and function with respect to detailed cytotoxic activity were unknown. The number of antibody drugs for veterinary use is limited, and ana-lyzing whether human antibody drugs bind to canine molecules would be of value since several cases have been reported of treating canine cancer with antibody drugs themselves or with single chain fragments of antibody drugs for humans that cross-react with canine molecules, such as CCR4 [18,19] and Her2 [20]. To date, only five anti-canine CD20 antibodies have been reported in the literature: clones 1E4 [10], 6C8 [12], NCD1.2 [13], 3N1 [14], and our 4E1-7-B_f [11]. Among

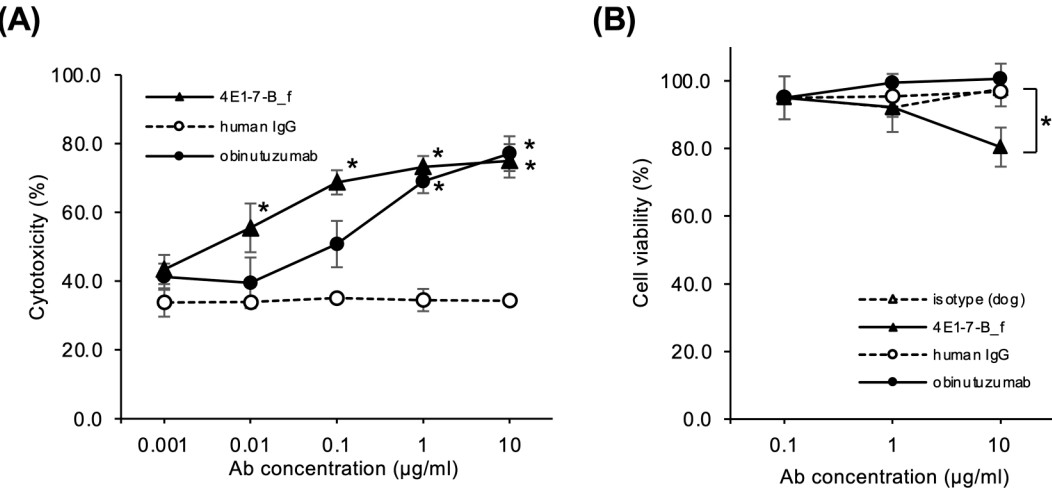

**Fig 3. *In vitro* functional analysis of 4E1-7-B_f and obinutuzumab biosimilar against canine lymphoma cell line, CLBL-1/luc.** (a) The ADCC assay was performed by incubating A20/cCD20/luc cells with serially diluted antibodies in the presence of effector cells, NK-92/cCD16γ. * indicates statistically significant difference (p < 0.05) between control human IgG antibody and 4E1-7-B_f or obinutuzumab by Tukey Kramer multiple comparison analysis. (b) The CDC assay was performed by incubating CLBL-1/luc cells with serially diluted antibody in the presence of rabbit complement. * indicates statistically significant difference (p < 0.05) between control dog IgG antibody and 4E1-7-B_f by Tukey Kramer multiple comparison analysis.

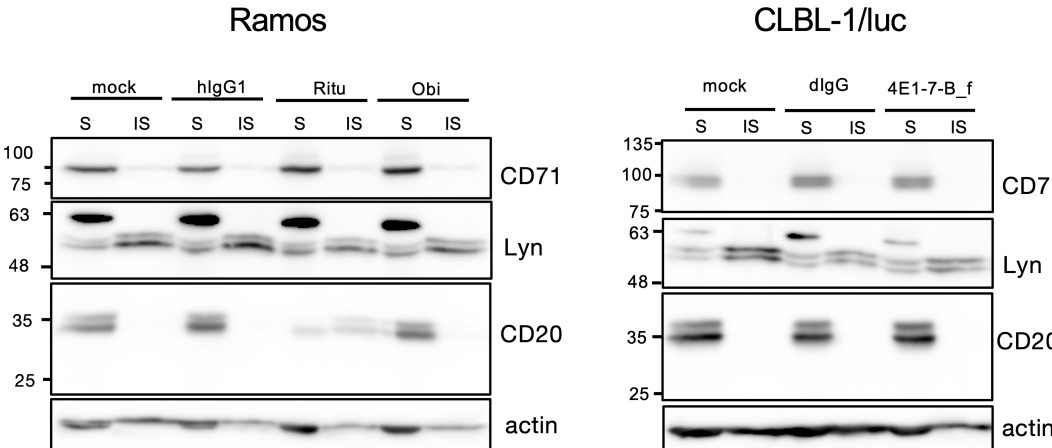

**Fig 4. Lipid raft translocation of CD20 in Ramos and CLBL-1/luc cells.** Ramos cells were stimulated with human IgG1, rituximab, or obinutuzumab biosimilar, and CLBL-1/luc cells were stimulated with dog IgG or 4E1-7-B_f for 5 min. After stimulation, soluble and insoluble fractions of cell lysates were extracted for western blotting with anti-CD71, anti-Lyn. and anti-CD20 antibody. CD71 and Lyn were used as controls for soluble and insoluble fractions, respectively. For transparency, the full-length images of the cropped western blots are shown in S4 Fig in S1 File. Representative results from two independent experiments are shown. Replicates of the full dataset are provided in S5 Fig in S1 File.

these, functional analyses have been conducted only for 1E4, 6C8, and our 4E1-7-B_f. The 1E4 clone is suggested to possess ADCC activity due to its binding to CD16; however, this has not been conclusively demonstrated through cellular assays [10]. Additionally, while it lacks CDC activity, it is reported to exhibit ADCP activity [21]. *In vivo* studies involving administration to healthy beagles [10] and canine B cell lymphoma cases [22] have shown a reduction in B cells. However, when compared to the marked B cell reduction observed with our 4E1-7-B_f antibody in beagles [11], the effect of 1E4 appears considerably weaker, suggesting that the strong ADCC activity demonstrated by 4E1-7-B_f plays a critical

role. Regarding clone 6C8, *in vitro* assays have shown that it possesses ADCP activity but lacks CDC activity and direct cell-killing ability [12]. However, its *in vivo* functionality has yet to be investigated. In this context, the 4E1-7-B_f antibody stands out not only for its previously reported strong ADCC activity [11] but also for the detailed epitope analysis and sub-class classification presented in this study. These findings provide significant insights into the antibody's potential therapeutic applications.

4E1-7-B_f showed no cross-reactivity with human, cat, or bovine CD20, indicating a highly specificity for canine CD20. However, the anti-human CD20 antibodies rituximab and the obinutuzumab biosimilar bound to canine CD20 overexpressed in NRK cells. Rituximab was reported not to bind to canine PBMCs [9], and this study also showed no binding to canine CD21+PBMCs, but unexpectedly bound to NRK cells overexpressing canine CD20. This is presumably because rituximab binds to CD20 overexpressed in NRK with a very weak binding affinity, whose $K_D$ value could not be calculated, and the expression of CD20 in PBMCs is too low to be detected. The obinutuzumab biosimilar bound not only to overexpressing cells but also to the CLBL-1/luc cell line and to canine CD21+B cells. Therefore, the $K_D$ of the obinutuzumab biosimilar was $17.42 \pm 1.76 \times 10^{-9}$ M, whereas the $K_D$ of rituximab was not measurable. This suggests that the obinutuzumab biosimilar has a higher affinity for canine CD20 than rituximab and may be able to detect endogenous canine CD20 if expressed at high levels. However, the binding was weaker than that of 4E1-7-B_f based on $K_D$ values. However, there are many other factors that can influence the apparent recognition and subsequent binding to the antigen-expressing cells, such as clustering, association with lipid rafts, potential internalization.

Regarding the ADCC activity, the obinutuzumab biosimilar showed approximately 100-fold less activity against A20/cCD20/luc cells than the 4E1-7-B_f antibody, although the obinutuzumab biosimilar is also a defucosylated antibody [7]. This result reflects the weak binding of the obinutuzumab biosimilar to canine CD20 as described above. Furthermore, although the 4E1-7-B_f antibody showed CDC activity at a concentration of 10 µg/mL, the obinutuzumab biosimilar did not show any CDC activity against CLBL-1/luc, consistent with obinutuzumab exhibiting weak CDC activity against human cells [16,23] and lower affinity to canine CD20. Thus, using obinutuzumab as a canine CD20-targeted therapy is unwise even if obinutuzumab can bind to canine CD20.

The epitope analysis of ofatumumab, rituximab, and obinutuzumab showed that each has a different epitope [17]. The 4E1-7-B antibody was produced by immunizing CD20 overexpressing cells, and the epitope of this antibody is unknown. Since our antibody does not cross-react with human CD20, we generated CD20 mutants based on the amino acid differences in CD20 between species and analyzed their epitopes. The M1 mutant completely lost the binding of 4E1-7-B_f, suggesting that the major epitope exists in this region while the M3, M4, and M5 mutants showed a slight decrease in binding, suggesting that changes in conformation due to these mutations may have affected binding.

Anti-CD20 antibodies are classified into type I and II antibodies based on differences in their functions such as CDC activity and CD20 localization to the lipid raft after antibody binding [5,6]. 4E1-7-B_f, like the obinutuzumab biosimilar, did not localize to the lipid raft in this study. Although CDC activity was observed at 10 µg/mL, we judged this to be a type II antibody like obinutuzumab because 4E1-7-B_f has strong ADCC activity [11]. Type II anti-CD20 antibodies, such as obinutuzumab, are known for sustained B-cell depletion in vivo, due to direct cytotoxicity and robust ADCC/ADCP activity despite weak CDC effects. This mechanism has been linked to prolonged therapeutic benefit in B-cell malignancies and autoimmune diseases [24]. Given its classification and potency, 4E1-7-B_f may offer similarly durable B-cell depletion in canine lymphoma, with potential clinical benefits such as prolonged disease control and reduced relapse. Further in vivo studies are needed to fully evaluate its clinical value.

The development of 4E1-7-B_f should be framed within the broader context of current veterinary oncology challenges, including the limited availability of species-specific antibody therapies and the need for more effective immunotherapeutic options in canine B-cell lymphoma. Our findings position 4E1-7-B_f as a promising therapeutic candidate that may address these unmet needs. The antibody demonstrated potent ADCC and CDC activity, underscoring its therapeutic potential. However, as seen in human medicine, immune evasion mechanisms may emerge-such as CD20

downregulation, selection of CD20-negative clones [25], immunosuppressive cytokine production [26], and upregulation of complement regulatory proteins such as CD55 and CD59 [27]. In addition, CD47-mediated resistance to phagocytosis may impair efficacy [28]. Therefore, combination therapies with CD47 or immune checkpoint inhibitors may enhance its clinical applicability.

Although we could not identify a clear epitope for the 4E1-7-B_f antibody from the results of this study, this is the first to classify it among anti-dog CD20 antibodies as a type II antibody. Alongside previous findings of strong ADCC and moderate CDC activity and complete elimination of peripheral blood B cells [11], this classification adds critical information for future clinical use. Taken together, our study not only provides new mechanistic insights into the function of 4E1-7-B_f but also addresses a critical gap in current veterinary oncology by offering a well-characterized, functionally potent, and potentially translatable antibody therapeutic for canine B-cell malignancies.

## Supporting information

**S1 File.   S1 Fig.** Gating strategy for low cytometry analysis. Debris were excluded, and lymphocytes included, using a forward scatter area (FSC) versus side scatter area (SSC) gate. Single cells were then selected on a FSC-A versus FSC-H plot to exclude signaling data from doublets. Dead cells were excluded based on propidium iodide staining and CD21+ and 4E1-7-B_f cells were shown. **S2 Fig.** Unedited and uncropped full image western blots, Round 1 and used in Fig 2 (c). Full western blot images for the expression of mutant forms of cCD20 (M1 to M5) in NRK cell lines. These westerns are the first of two replicates completed and were used to prepare Fig 2 (c). The order for each lane are: ladder, mock, WT, M1, M2, M3, M4 and M5. **S3 Fig.** Unedited and uncropped full image western blots, Round 2. Full western blot images for the expression of mutant forms of cCD20 (M1 to M5) in NRK cell lines. These westerns are the second of two replicates completed. The order for each lane are: ladder, mock, WT, M1, M2, M3, M4 and M5. **S4 Fig**. Unedited and uncropped full image western blots, Round 1 and used in Fig 4. Full western blot images for the expression of CD71, Lyn, CD20 and actin in soluble and insoluble fractions. These westerns are the first of two replicates completed and were used to prepare Fig 4. **S5 Fig**. Unedited and uncropped full image western blots, Round 2. Full western blot images for the expression of CD71, Lyn, CD20 and actin in soluble and insoluble fractions. These westerns are the second of two replicates completed.
(ZIP)

## Acknowledgments

We would like to acknowledge the technical expertise of the DNA Core Facility of the Center for Gene Research, Yamaguchi University. The authors would like to thank Mrs. K. Inoue and Dr. H. Ayame for technical assistance with the experiments.

## Author contributions

**Conceptualization:** Takuya Mizuno.

**Data curation:** Takuya Mizuno.

**Formal analysis:** Takuya Mizuno.

**Funding acquisition:** Takuya Mizuno.

**Investigation:** Takuya Mizuno.

**Project administration:** Takuya Mizuno.

**Resources:** Yukinari Kato.

**Supervision:** Takuya Mizuno.

**Validation:** Takuya Mizuno, Toshihiro Tsukui.

**Writing – original draft:** Takuya Mizuno.

**Writing – review & editing:** Toshihiro Tsukui, Masaya Igase.

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
