## [Decision Letter · Decision Letter 0]

PONE-D-24-19862Characterization of anti-canine CD20 antibody 4E1-7-B_f and comparison with commercially available anti-human CD20 antibodiesPLOS ONE

Dear Dr. Mizuno,

Thank you for submitting your manuscript to PLOS ONE. Firstly, we would like to apologize for the delay in processing your manuscript. It has been exceptionally difficult to secure reviewers to evaluate your study. We have now received one completed review, which is available below. The reviewer has raised significant scientific concerns about the study that need to be addressed in a revision.

Please note that we have only been able to secure a single reviewer to assess your manuscript. We are issuing a decision on your manuscript at this point to prevent further delays in the evaluation of your manuscript. Please be aware that the editor who handles your revised manuscript might find it necessary to invite additional reviewers to assess this work once the revised manuscript is submitted. However, we will aim to proceed on the basis of this single review if possible. 

We look forward to receiving your revised manuscript.

Kind regards,

Miquel Vall-llosera Camps

Senior Staff Editor

PLOS ONE

Journal Requirements:

"T. M. received research funding from Nippon Zenyaku Kogyo Co., Ltd. The remaining authors declare no conflicts of interest."

"T. M. received research funding from Nippon Zenyaku Kogyo Co., Ltd. The remaining authors declare no conflicts of interest."

4. In this instance it seems there may be acceptable restrictions in place that prevent the public sharing of your minimal data. However, in line with our goal of ensuring long-term data availability to all interested researchers, PLOS’ Data Policy states that authors cannot be the sole named individuals responsible for ensuring data access (http://journals.plos.org/plosone/s/data-availability#loc-acceptable-data-sharing-methods).

Reviewers' comments:

Reviewer's Responses to Questions

**Comments to the Author**

1. Is the manuscript technically sound, and do the data support the conclusions?

Reviewer #1: Partly

2. Has the statistical analysis been performed appropriately and rigorously? 

Reviewer #1: Yes

3. Have the authors made all data underlying the findings in their manuscript fully available?

Reviewer #1: Yes

4. Is the manuscript presented in an intelligible fashion and written in standard English?

Reviewer #1: Yes

5. Review Comments to the Author

Reviewer #1: In the present manuscript, the authors present a sequel on their research on an anti-canine CD20 antibody, which has a significant deleterious effect on CD20-positive cells, showing an ADCC activity and CDC activity (the latter at high concentration of 10 micrograms per mL). Based on its biological activity and the localization to lipid rafts, the authors classify the test antibody as a Type II antibody. In all tests, they compare it with rituximab and glycoengineered obinutuzumab. The experiments they use are well designed and several important controls (isotypes) are used. Certainly, the discovery of novel therapeutic reagents that could combat the wide-spread serious condition of canine lymphoma is a valuable study topic of high translational value.

The delineation of the relevant epitope is also performed, with cell-expressed CD20 mutants that appear of the same expression using a western blot. Interestingly, one of the mutants is irresponsive to the antibody, while few others are less responsive, which is interpreted with a conformational change of the membrane-bound antigen. The expression of the membrane-bound mutant antigen is controlled with an anti-FLAG tag antibody, which might be the reason why the differences in the conformation remained unobserved. The titrations of other anti-CD20 antibodies, such as rituximab, obinutuzumab (or biosimilar), and ofatumumab, as well as the test antibodies could give more insights on the conformational state (and accessibility) of the CD20 mutants used for this study.

Several comments of the authors refer to the antigen affinity of the novel antibody, but at present there are no measurements presented, either titrations or other means of evaluation of the strength of the interaction. This data set should be presented – the manuscript would profit from the exact delineation of the antibody activity in relation to the number of the antigen molecules displayed on the cell surface (especially as the authors use different target cell types to measure the antibody reactivity).

Further, the authors cite only 13 references, which should be expanded, and this should not be difficult for the therapeutically attractive antigen CD20 which is in the centre of this research contribution.

Finally, the conclusion could be oriented into the development of the studied antibody for therapeutic purposes – seeing its activity, probably affinity maturation would be helpful, and these are challenging to design for CD20. Are there data on the expression level, monomeric state, maximal solubility, and stability of the antibody?

Please find below a list of remarks which I hope you will find helpful.

Line 34: “classified this antibody as a type II antibody similar to the obinutuzumab” – this statement is valuable only to the CD20-specialized audience, I propose to complement it by describing the functional particularities of this antibody class

Line 46: “Cytotoxic antibody drugs such as anti-CD20 antibodies have antitumor

effects through antibody-dependent cell cytotoxicity (ADCC), complement-dependent

cytotoxicity (CDC), antibody-dependent cell phagocytosis (ADCP), and direct cytotoxic

activity, but these functions vary widely among individual antibody drugs” – this statement should be reworded to point specifically at anti-CD20 antibodies: Anti-CD20 antibodies are cytotoxic antibody drugs exerting antitumor effects…, and concluding in: these functions vary widely among the therapeutic agents directed towards this target, or similar.

Line 58: their efficacies

Line 62: “no anti-canine CD20 antibody drug is available” – approved? Commercially available?

Line 63: “the functional mechanism“ – mode or mechanism of action

Line 66: based on this molecule.

Line 78: “on the localization of the lipid raft of CD20“ – based on the CD20 localization into lipid rafts?

Line 126: what precisely is FACS buffer composed of?

Line 170: using

Line 177: overexpressed

Line 179: overexpressing

Materials and methods: all antibodies should be listed with RRIDs. Isotype controls used for FACS experiments are not mentioned.

Line 206 (and Figure 1C): what is the population reacting with dog isotype control? Were the dead cells gated out?

Line 259: Measure and extent of statistical significance should be described in the Figure legend.

Line 282: “The function of individual cytotoxic antibody drugs is important” – this sentence is very generic, it would be better to reword, that it is crucial to define the mode of action of different cytotoxic antibodies to design their further development, or similar

Line 298: The comments on affinity, without actual measurements directed in this way, not even titrations of the positive cells, are not well founded at this stage. There are many other factors that can influence the apparent recognition and subsequent binding to the antigen-expressing cells, such as clustering, association with lipid rafts, potential internalization – please consider these in the comments.

Line 306: something wrong with the micro (Greek mu) sign

Paragraph starting with line 301: as mentioned before, it is difficult to judge the affinity based on the reactivity with cell-bound antigens, especially in the case of tetraspanin molecules such as CD20, which show certain patterns of clustering and mobilization in response to antibody treatment.

Line 311: the reference of Klein et al. is not in the correct format

Line 322: something wrong with the micro (Greek mu) sign

Line 327: “is the first antibody that belongs to type II“ – a brief overview of the findings on anti-canine CD20 antibodies would be helpful (although the authors present one in the reference 5).

6. PLOS authors have the option to publish the peer review history of their article (what does this mean? ). If published, this will include your full peer review and any attached files.

**Do you want your identity to be public for this peer review?** For information about this choice, including consent withdrawal, please see our Privacy Policy .

Reviewer #1: No

---

## [Author Response · Author response to Decision Letter 1]

13 Feb 2025

We have modified it to match PLOS ONE's style requirements as you indicated.

"T. M. received research funding from Nippon Zenyaku Kogyo Co., Ltd. The remaining authors declare no conflicts of interest."

Employees of Nippon Zenyaku Kogyo Co., Ltd. are included as co-authors, and their roles are described in the Author contributions.

"T. M. received research funding from Nippon Zenyaku Kogyo Co., Ltd. The remaining authors declare no conflicts of interest."

This information was added into the cover letter.

4. In this instance it seems there may be acceptable restrictions in place that prevent the public sharing of your minimal data. However, in line with our goal of ensuring long-term data availability to all interested researchers, PLOS’ Data Policy states that authors cannot be the sole named individuals responsible for ensuring data access (http://journals.plos.org/plosone/s/data-availability#loc-acceptable-data-sharing-methods).

We don't have a system to manage such third-party data, so how should we solve this?

I attached them in my previous submission.

Reviewers' comments:

Reviewer #1: In the present manuscript, the authors present a sequel on their research on an anti-canine CD20 antibody, which has a significant deleterious effect on CD20-positive cells, showing an ADCC activity and CDC activity (the latter at high concentration of 10 micrograms per mL). Based on its biological activity and the localization to lipid rafts, the authors classify the test antibody as a Type II antibody. In all tests, they compare it with rituximab and glycoengineered obinutuzumab. The experiments they use are well designed and several important controls (isotypes) are used. Certainly, the discovery of novel therapeutic reagents that could combat the wide-spread serious condition of canine lymphoma is a valuable study topic of high translational value.

The delineation of the relevant epitope is also performed, with cell-expressed CD20 mutants that appear of the same expression using a western blot. Interestingly, one of the mutants is irresponsive to the antibody, while few others are less responsive, which is interpreted with a conformational change of the membrane-bound antigen. The expression of the membrane-bound mutant antigen is controlled with an anti-FLAG tag antibody, which might be the reason why the differences in the conformation remained unobserved. The titrations of other anti-CD20 antibodies, such as rituximab, obinutuzumab (or biosimilar), and ofatumumab, as well as the test antibodies could give more insights on the conformational state (and accessibility) of the CD20 mutants used for this study.

Several comments of the authors refer to the antigen affinity of the novel antibody, but at present there are no measurements presented, either titrations or other means of evaluation of the strength of the interaction. This data set should be presented – the manuscript would profit from the exact delineation of the antibody activity in relation to the number of the antigen molecules displayed on the cell surface (especially as the authors use different target cell types to measure the antibody reactivity).

Further, the authors cite only 13 references, which should be expanded, and this should not be difficult for the therapeutically attractive antigen CD20 which is in the centre of this research contribution.

Finally, the conclusion could be oriented into the development of the studied antibody for therapeutic purposes – seeing its activity, probably affinity maturation would be helpful, and these are challenging to design for CD20. Are there data on the expression level, monomeric state, maximal solubility, and stability of the antibody?

To evaluate the affinity of obinutuzumab and 4E1-7-B_f antibodies for NRK/cCD20, we performed a flow cytometry assay using a dilution series of the antibodies and compared the KD values. Similarly, we also performed an assay using CD20 mutants-expressing cells using a dilution series of the 4E1-7-B_f antibody to evaluate their affinity. The data are shown in Figure 2D.

As you indicated, since the number of references cited was small, the revised manuscript cites the newly required references, bringing the total to 21.

We are developing this antibody as a therapeutic animal antibody drug, and in the process, we are studying the expression level, monomeric state, maximum solubility, and stability of the antibody, which are confidential information held by the animal drug company and are difficult to disclose here.

Please find below a list of remarks which I hope you will find helpful.

Line 34: „classified this antibody as a type II antibody similar to the obinutuzumab” – this statement is valuable only to the CD20-specialized audience, I propose to complement it by describing the functional particularities of this antibody class

According to your suggestion, I described the type II antibody in Abstract (Lines 35-37).

Line 46: “Cytotoxic antibody drugs such as anti-CD20 antibodies have antitumor effects through antibody-dependent cell cytotoxicity (ADCC), complement-dependent cytotoxicity (CDC), antibody-dependent cell phagocytosis (ADCP), and direct cytotoxic activity, but these functions vary widely among individual antibody drugs” – this statement should be reworded to point specifically at anti-CD20 antibodies: Anti-CD20 antibodies are cytotoxic antibody drugs exerting antitumor effects…, and concluding in: these functions vary widely among the therapeutic agents directed towards this target, or similar.　　

According to your suggestion, I modified the sentence (Lines 48-51).

Line 58: their efficacies

Line 62: “no anti-canine CD20 antibody drug is available” – approved? Commercially available?

Line 63: “the functional mechanism“ – mode or mechanism of action

Line 66: based on this molecule.

Line 78: “on the localization of the lipid raft of CD20“ – based on the CD20 localization into lipid rafts?

Line 126: what precisely is FACS buffer composed of?

Line 170: using

Line 177: overexpressed

Line 179: overexpressing

Thank you for your detailed corrections. All have been corrected as you indicated.

Materials and methods: all antibodies should be listed with RRIDs. Isotype controls used for FACS experiments are not mentioned.

I have corrected those as you indicated (Lines 150-199).

Line 206 (and Figure 1C): what is the population reacting with dog isotype control? Were the dead cells gated out?

Yes, the dead cells were gated out by PI staining, which is described in the revised manuscript (Lines 158-159). The population reacting with dog isotype control was Fc-mediated non specific binding of dog IgG isotype control in CD21+ fraction, but we do not know what is the population in CD21- fraction. However, important point is there is no difference between dog IgG binding and 4E1-7-B_f reaction in CD21- faction, but there are increase of binding of 4E1-7-B_f in CD21+ fraction.

Line 259: Measure and extent of statistical significance should be described in the Figure legend.

I have filled in the necessary areas in Figs 2 and 3.

Line 282: “The function of individual cytotoxic antibody drugs is important” – this sentence is very generic, it would be better to reword, that it is crucial to define the mode of action of different cytotoxic antibodies to design their further development, or similar

I have corrected it as you pointed out (Lines 352-353).

Line 298: The comments on affinity, without actual measurements directed in this way, not even titrations of the positive cells, are not well founded at this stage. There are many other factors that can influence the apparent recognition and subsequent binding to the antigen-expressing cells, such as clustering, association with lipid rafts, potential internalization – please consider these in the comments.　　

The revised manual lists the KD value measured using flow cytometry, but we have also mentioned the possibility you pointed out in lines 374-377.

Line 306: something wrong with the micro (Greek mu) sign

I have corrected.

Paragraph starting with line 301: as mentioned before, it is difficult to judge the affinity based on the reactivity with cell-bound antigens, especially in the case of tetraspanin molecules such as CD20, which show certain patterns of clustering and mobilization in response to antibody treatment.　

As mentioned above, explanations have been added where necessary.

Line 311: the reference of Klein et al. is not in the correct format

I have corrected.

Line 322: something wrong with the micro (Greek mu) sign

I have corrected.

Line 327: “is the first antibody that belongs to type II“ – a brief overview of the findings on anti-canine CD20 antibodies would be helpful (although the authors present one in the reference 5).

I have corrected.

---

## [Decision Letter · Decision Letter 1]

PONE-D-24-19862R1Characterization of anti-canine CD20 antibody 4E1-7-B_f and comparison with commercially available anti-human CD20 antibodiesPLOS ONE

Dear Dr. Mizuno,

Thank you for submitting your manuscript to PLOS ONE. After careful consideration, we feel that it has merit but does not fully meet PLOS ONE’s publication criteria as it currently stands. Therefore, we invite you to submit a revised version of the manuscript that addresses the points raised during the review process.

We look forward to receiving your revised manuscript.

Kind regards,

Cho-Hao Howard Lee, M.D.

Academic Editor

PLOS ONE

Journal Requirements:

Reviewers' comments:

Reviewer's Responses to Questions

**Comments to the Author**

1. If the authors have adequately addressed your comments raised in a previous round of review and you feel that this manuscript is now acceptable for publication, you may indicate that here to bypass the “Comments to the Author” section, enter your conflict of interest statement in the “Confidential to Editor” section, and submit your "Accept" recommendation.

Reviewer #1: All comments have been addressed

Reviewer #2: All comments have been addressed

Reviewer #3: (No Response)

Reviewer #4: (No Response)

2. Is the manuscript technically sound, and do the data support the conclusions?

Reviewer #1: Partly

Reviewer #2: Yes

Reviewer #3: Partly

Reviewer #4: Partly

3. Has the statistical analysis been performed appropriately and rigorously? 

Reviewer #1: Yes

Reviewer #2: Yes

Reviewer #3: Yes

Reviewer #4: Yes

4. Have the authors made all data underlying the findings in their manuscript fully available?

Reviewer #1: Yes

Reviewer #2: Yes

Reviewer #3: Yes

Reviewer #4: Yes

5. Is the manuscript presented in an intelligible fashion and written in standard English?

Reviewer #1: Yes

Reviewer #2: Yes

Reviewer #3: Yes

Reviewer #4: Yes

6. Review Comments to the Author

Reviewer #1: In the revised version of the article, the authors have added additional data showing the dose-response titrations of their antibody and other tested antibodies and improved on the description of materials and as well as the style of data presentation. They have also included more references. Unfortunately, they have not included biophysical data on their antibody but have explained why this is not possible. Please find a short list of remarks below.

Line 51: ofatumumab

Line 101: San Diego

Lines 107 and 111: the sequence of the FLAG-tag used here is probably DYKDHDID – this is different from the conventional FLAG sequence, please mention this fact here.

Line 321: this suggests

Line 337: each has a different epitope

Figure 2D: unfortunately, it is not indicated which colour correspond to which dilution, please correct. Additionally, certain mutants are only measured with 3 concentrations of the antibody (3 dots), while others have several? A plot showing dose-response (concentration on x, signal of staining on y axis) should be added here, and EC50 could be added as a figure in the plot or in the text.

Epitope mapping experiment is very elegant, but would it be possible to test the mutants for the binding of ofatumumab (M1, 2 and 3) or RX (M4 and 5) and Obinutuzumab (M5) – this could be informative on retaining the CD20 conformation.

Reviewer #2: This manuscript is a revised version (I wasn’t involved in the first-round peer-review), overall, they did a great job on responding the reviewer.

I have one minor question:

Fig 3A, isn’t the viability higher in 4E1-7B_f group? It means lower killing of the NK effectors cells, which suggests weak ADCC in the 4E1-7B_f antibody, right? This is the opposite of author’s claim, so please clarify.

Reviewer #3: Merits

①Scientific Significance

a. The study provides valuable translational insights into the potential therapeutic use of 4E1-7-B_f in canine lymphoma treatment.

b. Comparative analysis with rituximab and an obinutuzumab biosimilar strengthens the manuscript’s impact in veterinary oncology.

②Methodological Strengths

a. The comprehensive epitope analysis using mutant CD20 constructs adds depth to the understanding of antibody-antigen interactions.

b. The study effectively employs flow cytometry-based affinity assays and functional cytotoxicity assays (ADCC, CDC), enhancing the robustness of its conclusions.

③Novel Findings

a. The study demonstrates that 4E1-7-B_f exhibits stronger ADCC and CDC activity than the obinutuzumab biosimilar, supporting its therapeutic potential.

b. The classification of 4E1-7-B_f as a Type II anti-CD20 antibody, distinct from rituximab, provides important mechanistic insights.

④Clinical Relevance

This study bridges the gap between basic research and clinical application, suggesting 4E1-7-B_f as a candidate for therapeutic development.

Areas for Improvement:

①Figures and Data Presentation

a. Figure 1C: The labeling of CD21+ B cells and gating strategy should be clarified to ensure accurate interpretation.

b. Figure 2D: The statistical significance of binding affinities (KD values) should be explicitly stated in the figure legend.

c. Western blot images (Figure 4): The band intensities should be quantified to provide a more objective comparison of lipid raft translocation.

②Clarity in Statistical Methods

a. The p-values and statistical test details should be explicitly mentioned for each figure.

b. If multiple comparisons were performed, corrections for multiple testing should be addressed.

③Expansion of Discussion

a, The study should compare 4E1-7-B_f with other reported anti-canine CD20 antibodies beyond rituximab and obinutuzumab.

b. The discussion should acknowledge potential in vivo challenges, such as immune evasion mechanisms in a clinical setting.

c. The therapeutic implications of a Type II CD20 antibody in long-term B-cell depletion should be elaborated.

④References and Literature Review

The manuscript currently cites only 21 references, which is limited for a study of this scope. Additional literature on anti-CD20 antibody therapy in veterinary oncology should be included.

Suggested Revisions

①Improve Figure Legends: Ensure statistical annotations and experimental details are fully described.

②Enhance Discussion: Expand on clinical implications, potential limitations, and future research directions.

③Clarify Statistical Methods: Explicitly report p-values, statistical tests used, and any multiple comparison adjustments.

④Increase Reference Count: Incorporate additional relevant studies on anti-canine CD20 therapy.

Reviewer #4: This study provides a comprehensive functional and mechanistic analysis of the novel anti-canine CD20 monoclonal antibody 4E1-7-B_f, positioning it as a candidate therapeutic for canine lymphoma. The work demonstrates that 4E1-7-B_f exhibits strict species specificity for canine CD20, with no cross-reactivity to human, feline, or bovine homologs. While the obinutuzumab biosimilar showed weak binding to canine CD20 in engineered cell lines and primary B cells, 4E1-7-B_f achieved superior binding affinity and significantly stronger functional activity—100-fold greater ADCC potency and detectable CDC effects at 10 µg/mL, unlike the biosimilar. Epitope mapping identified critical residues in the extracellular domain (mutant M1) essential for binding, though conformational dependencies were suggested by reduced affinity in M3/M4 mutants. Lipid raft localization experiments classified 4E1-7-B_f as a type II antibody akin to obinutuzumab, contrasting with type I rituximab. The study bridges veterinary and human immunotherapy paradigms, offering the first classification of a canine-specific type II anti-CD20 agent while addressing translational gaps in veterinary oncology.

Revision Recommendations

1. Issue: Methods sections lack critical details (e.g., FACS buffer composition, statistical thresholds). Detailed methods ensure reproducibility

Revision:

Add precise technical details in Methods (e.g., FACS buffer recipe, criteria for excluding dead cells via PI staining).

2. Issue: The introduction/discussion underemphasizes the unmet need in canine lymphoma (e.g., chemo resistance rates) and recent advances in veterinary immunotherapy.

Revision:

Add statistics on chemo failure rates in canine B-cell lymphoma. Cite 2023–2024 studies on anti-CCR4/Her2 therapies in dogs. Framing 4E1-7-B_f within current veterinary oncology challenges and solutions highlights its translational significance.

3. Issue: Type II antibodies (like obinutuzumab) typically have weak CDC activity, yet 4E1-7-B_f shows moderate CDC at 10 µg/mL. This discrepancy isn’t explained.

Revision:

Explain this discrepancy in the discussion part.

7. PLOS authors have the option to publish the peer review history of their article (what does this mean? ). If published, this will include your full peer review and any attached files.

**Do you want your identity to be public for this peer review?** For information about this choice, including consent withdrawal, please see our Privacy Policy .

Reviewer #1: No

Reviewer #2: No

Reviewer #3: **Yes: ** Xiaoyi Zhang, MD

Reviewer #4: No

---

## [Author Response · Author response to Decision Letter 2]

2 Apr 2025

Dear Editor and Reviewers,

We would like to thank you for your careful review of our manuscript entitled " Characterization of anti-canine CD20 antibody 4E1-7-B_f and comparison with commercially available anti-human CD20 antibodies" (Manuscript ID: PONE-D-24-19862R1). We sincerely appreciate the insightful and constructive comments provided by the reviewers and the editor, which have helped us to improve the quality and clarity of our work.

We have carefully addressed all the comments and suggestions, and have revised the manuscript accordingly. Below, we provide a detailed, point-by-point response to each of the reviewers’ comments. For clarity, the reviewers’ comments are shown in italic, and our responses are provided below each comment.

We hope that the revised version of the manuscript addresses all concerns and meets the requirements for publication in Plos One.

Sincerely,

Takuya Mizuno (on behalf of all authors)

Response to Reviewer #1:

Line 51: ofatumumab

Line 101: San Diego

Lines 107 and 111: the sequence of the FLAG-tag used here is probably DYKDHDID – this is different from the conventional FLAG sequence, please mention this fact here.

Line 321: this suggests

Line 337: each has a different epitope

We have corrected those as you indicated.

Figure 2D: unfortunately, it is not indicated which colour correspond to which dilution, please correct. Additionally, certain mutants are only measured with 3 concentrations of the antibody (3 dots), while others have several? A plot showing dose-response (concentration on x, signal of staining on y axis) should be added here, and EC50 could be added as a figure in the plot or in the text.

The y-axis of this graph represents the Kd values, and as shown in the M&M they are calculated from the dilution columns of the individual antibodies, so the different colors indicate each Kd value obtained from different independent experiments. The number of dots varies from antibody to antibody because the number of trials for each independent experiment is different. The left plot in Figure 2D has been changed to a plot showing dose-response, as you suggested. Since the Kd values are shown in the graph, which we believe can be evaluated as an alternative to EC50. We have added more details about these in Figure legend (lines 274-279 in the revised manuscript with track changes).

Epitope mapping experiment is very elegant, but would it be possible to test the mutants for the binding of ofatumumab (M1, 2 and 3) or RX (M4 and 5) and Obinutuzumab (M5) – this could be informative on retaining the CD20 conformation.

I have conducted the experiment you suggested on rituximab and obinutuzumab. The results are shown in right figure, and we actually found that he M5 mutant was found to be significantly increased in reactivity by both antibodies. These facts, however, are not the subject of our present manuscript and are therefore not included in the text.

-

Response to Reviewer #2: This manuscript is a revised version (I wasn’t involved in the first-round peer-review), overall, they did a great job on responding the reviewer.

I have one minor question:

Fig 3A, isn’t the viability higher in 4E1-7B_f group? It means lower killing of the NK effectors cells, which suggests weak ADCC in the 4E1-7B_f antibody, right? This is the opposite of author’s claim, so please clarify.

Thank you for your important point, the label on the Y-axis in Figure 3A is incorrect and represents cell injury induced by ADCC, so we have corrected the label on the Y-axis of the graph as such.

Response to Reviewer #3: Merits

Areas for Improvement:

①Figures and Data Presentation

a. Figure 1C: The labeling of CD21+ B cells and gating strategy should be clarified to ensure accurate interpretation.

We added a description of the gating strategy as Supplement Figure1.

b. Figure 2D: The statistical significance of binding affinities (KD values) should be explicitly stated in the figure legend.

As you pointed out, I have explained it in the Figure legend (Lines 270-279 in the revised manuscript with track changes).

c. Western blot images (Figure 4): The band intensities should be quantified to provide a more objective comparison of lipid raft translocation.

We have stated that when CLBL-1/luc is treated with 4E1-7-B_f, CD20 does not migrate to the lipid raft. since it is obvious in western blotting that no CD20 band is detected in the IS fraction. We do not believe that reading the intensity would change the results.

②Clarity in Statistical Methods

a. The p-values and statistical test details should be explicitly mentioned for each figure.

b. If multiple comparisons were performed, corrections for multiple testing should be addressed.

We have revised the figure legends to explicitly mention the p-values and details of the statistical tests used for each figure. Additionally, where multiple comparisons were performed, we have addressed corrections for multiple testing accordingly.

③Expansion of Discussion

a, The study should compare 4E1-7-B_f with other reported anti-canine CD20 antibodies beyond rituximab and obinutuzumab.

There are very limited numbers of published papers about anti-canine CD20 antibodies, but we mentioned about this in lines 338-354 in discussion section of the revised manuscript with track changes.

b. The discussion should acknowledge potential in vivo challenges, such as immune evasion mechanisms in a clinical setting.

We mentioned about this in lines 404-415 in discussion section of the revised manuscript with track changes.

c. The therapeutic implications of a Type II CD20 antibody in long-term B-cell depletion should be elaborated.

We mentioned about this in lines 395-403 in discussion section of the revised manuscript with track changes.

④References and Literature Review

The manuscript currently cites only 21 references, which is limited for a study of this scope. Additional literature on anti-CD20 antibody therapy in veterinary oncology should be included.

We thank the reviewer for this valuable suggestion. In response, we have reviewed the relevant literature more thoroughly and have expanded the reference list to include additional studies related to anti-CD20 antibody therapy. Specifically, we have added references discussing the development and clinical evaluation of anti-canine CD20 antibodies, cross-species reactivity of therapeutic antibodies, and immune evasion mechanisms relevant to antibody therapy. As a result, the number of references has increased from 21 to 28, and the discussion section has been revised accordingly to better place our findings within the context of existing research.

Response to Reviewer #4:

Revision Recommendations

1. Issue: Methods sections lack critical details (e.g., FACS buffer composition, statistical thresholds). Detailed methods ensure reproducibility

Revision:

Add precise technical details in Methods (e.g., FACS buffer recipe, criteria for excluding dead cells via PI staining).

The recipe for the FACS buffer is given on line 137, and the criteria are given on line 149.

2. Issue: The introduction/discussion underemphasizes the unmet need in canine lymphoma (e.g., chemo resistance rates) and recent advances in veterinary immunotherapy.

Revision:

Add statistics on chemo failure rates in canine B-cell lymphoma. Cite 2023–2024 studies on anti-CCR4/Her2 therapies in dogs. Framing 4E1-7-B_f within current veterinary oncology challenges and solutions highlights its translational significance.

I added line 62-67 regarding “Add statistics on chemo failure rates in canine B-cell lymphoma. Regarding “Cite 2023-2024 studies on anti-CCR4/Her2 therapies in dogs.”, I didn't add it because it has nothing to do with the current study. I also added lines 410-415 and 434-437 in discussion section of the revised manuscript with track changes, regarding “Framing 4E1-7-B_f within current veterinary oncology challenges and solutions highlights its translational significance.”

3. Issue: Type II antibodies (like obinutuzumab) typically have weak CDC activity, yet 4E1-7-B_f shows moderate CDC at 10 µg/mL. This discrepancy isn’t explained.

Revision:

Explain this discrepancy in the discussion part.

We have not been able to explain this well: type II antibodies, such as obinutuzumab, have less CDC activity than type I antibodies, but not no CDC activity at all. The fact that our study failed to show CDC activity with obinutuzumab at 10µg/ml and our antibody showed CDC activity does not mean that our antibody has higher CDC activity than obinutuzumab, but rather reflects its affinity for canine CD20. We believe that this is a reflection of the affinity of our antibody for canine CD20. I mentioned this a bit in the discussion section to avoid any misunderstanding by the reader.

---

## [Decision Letter · Decision Letter 2]

Characterization of anti-canine CD20 antibody 4E1-7-B_f and comparison with commercially available anti-human CD20 antibodies

PONE-D-24-19862R2

Dear Dr. Takuya Mizuno,

We’re pleased to inform you that your manuscript has been judged scientifically suitable for publication and will be formally accepted for publication once it meets all outstanding technical requirements.

Kind regards,

Cho-Hao Howard Lee, M.D.

Academic Editor

PLOS ONE

---

## [Editor Report · Acceptance letter]

PONE-D-24-19862R2

PLOS ONE

Dear Dr. Mizuno,

I'm pleased to inform you that your manuscript has been deemed suitable for publication in PLOS ONE. Congratulations! Your manuscript is now being handed over to our production team.

Kind regards,

on behalf of

Dr. Cho-Hao Howard Lee

Academic Editor

PLOS ONE